# The Beneficial Impact of a Novel Pancreatic Polypeptide Analogue on Islet Cell Lineage

**DOI:** 10.3390/ijms26094215

**Published:** 2025-04-29

**Authors:** Wuyun Zhu, Neil Tanday, Peter R. Flatt, Nigel Irwin

**Affiliations:** Diabetes Research Centre, Schools of Biomedical Science and Pharmacy, Ulster University, Co. Londonderry, Coleraine BT52 1SA, UKn.tanday@ulster.ac.uk (N.T.); pr.flatt@ulster.ac.uk (P.R.F.)

**Keywords:** pancreatic polypeptide, NPY4R, cell lineage, transdifferentiation

## Abstract

(Proline3)PP, or (P^3^)PP, is an enzymatically stable, neuropeptide Y4 receptor (NPY4R)-selective, pancreatic polypeptide (PP) analogue with established weight-lowering and pancreatic islet morphology benefits in obesity-diabetes. In the current study, we now investigate the impact of twice-daily (P^3^)PP administration (25 nmol/kg) for 11 days on islet cell lineage, using streptozotocin (STZ) diabetic Ins1*^Cre/+^*;Rosa26-eYFP and Glu*^CreERT2^*;Rosa26-eYFP transgenic mice with enhanced yellow fluorescent protein (eYFP) labelling of beta-cell and alpha-cells, respectively. (P^3^)PP had no obvious impact on body weight or blood glucose levels in STZ-diabetic mice at the dose tested, but did return food intake towards control levels in Ins1*^Cre/+;^*Rosa26-eYFP mice. Notably, pancreatic insulin content was augmented by (P^3^)PP treatment in both Ins1*^Cre/+;^*Rosa26-eYFP and Glu*^CreERT2^*;Rosa26-eYFP mice, alongside enhanced beta-cell area and reduced alpha-cell area. Beneficial (P^3^)PP-induced changes on islet morphology were consistently associated with decreased beta-cell apoptosis, while (P^3^)PP also augmented beta-cell proliferation in Ins1*^Cre/+;^*Rosa26-eYFP mice. Alpha-cell turnover rates were returned towards healthy control levels by (P^3^)PP intervention in both mouse models. In terms of islet cell lineage, increased transition of alpha- to beta-cells as well as decreased beta- to alpha-cell differentiation were shown to contribute towards the enhancement of beta-cell area in (P^3^)PP-treated mice. Together these data reveal, for the first time, sustained NPY4R activation positively modulates beta-cell turnover, as well as islet cell plasticity, to help preserve pancreatic islet architecture following STZ-induced metabolic stress.

## 1. Introduction

The preservation and/or regeneration of pancreatic beta-cell mass represents an attractive therapeutic approach for diabetes [1]. In this regard, activation of neuropeptide Y4 receptors (NPY4Rs) by pancreatic polypeptide (PP) has previously been demonstrated to protect beta-cells against streptozotocin (STZ)-induced apoptosis [2], with the long-acting enzyme-resistant PP analogue, (Proline3)PP or (P^3^)PP, recapitulating these benefits [3]. Accordingly, we have recently demonstrated that sustained (P^3)^PP administration exerts positive effects on islet cell turnover, leading to augmented beta-cell mass and pancreatic insulin content in a rodent model of obesity-diabetes [4]. Typically, beta-cell loss in diabetes has been attributed to stress-induced apoptosis, as a consequence of the relatively poor antioxidant defence capacity of these cells in the face of persistent hyperglycaemia [5]. However, levels of beta-cell apoptosis in human type 2 diabetes mellitus (T2DM) are lower than expected [6], suggesting the intrinsic plasticity of pancreatic endocrine cells may also be fundamentally important for regulating beta-cell mass and function [7].

Intriguingly, PP-producing islet gamma-cells have been identified as a potential source for beta-cell regeneration in both rodents and humans, with these cells shown to be capable of activating insulin stimulus-secretion coupling pathways following beta-cell ablation [8]. This change in cellular phenotype is entirely credible when considering the inherent heterogeneity of beta-cells, with some of these cells being of Ppy lineage, the gene that encodes for PP [9]. Furthermore, transcriptomic analysis reveals down-regulation of Ppy messenger ribonucleic acid (mRNA) following induction of islet dysfunction in rodents [10]. In addition, islet benefits of bariatric surgery have recently been linked to alterations of Ppy positive cell transdifferentiation events [11]. Thus, it is conceivable that the previously reported benefits of (P^3^)PP of pancreatic islet morphology [4] could be linked to positive alterations of islet cell lineage.

Therefore, in the present study we have employed streptozotocin (STZ)-induced diabetic transgenic mice with alpha- and beta-cell lineage tracing capabilities, to investigate the direct impact sustained (P^3^)PP treatment on islet cell plasticity. The primary aim being to uncover potential mechanisms underlying previously reported (P^3^)PP-induced benefits on pancreatic islet morphology and overall metabolism [4]. As such, fully characterised STZ insulin-deficient Ins1*^Cre/+^*;Rosa26-eYFP and Glu*^CreERT2^*;Rosa26-eYFP mice were utilised [5,12], with subsequent twice-daily (P^3^)PP administration for 11 days, permitting exploration of effects on both alpha- and beta-cell de- and trans-differentiation in the setting of multiple low-dose STZ-related beta-cell loss. Importantly, (P^3^)PP treatment was initiated two days prior to STZ injections to probe beta-cell protective actions, with peptide treatment continued for a further six days post STZ. As such, unlike high-dose STZ where an almost total beta-cell ablation occurs within 24 h, the multiple-low-dose STZ murine model mimics beta-cell changes in human diabetes such as islet cell immune infiltration and beta-cell dysfunction [13]. A gradual loss of beta-cell mass is observed, linked to lymphocytic infiltration of islet cells that peaks around day 11 [14]. This corresponds directly to the final day of the current experiment, reinforcing the concept that the current (P3)PP treatment regimen was employed as a preventive intervention. Collectively, data from the transgenic mouse models confirm that islet NPY4R modulation is linked to encouraging effects on beta-cell turnover and islet cell plasticity, resulting in notable benefits to overall pancreatic islet morphology.

## 2. Results

### 2.1. Effects (P^3^)PP on Body Weight, Food Intake, Blood Glucose and Pancreatic Insulin Content in STZ-Diabetic Ins1^Cre/+^;Rosa26-eYFP and Glu^CreERT2^;Rosa26-eYFP Mice

STZ reduced body weight, and subsequently increased percentage body weight loss, in both Ins1*^Cre/+^*;Rosa26-eYFP and Glu*^CreERT2^*;Rosa26-eYFP mice (Figure 1A,B and Figure 2A,B), but effects were only significant in Ins1*^Cre/+^*;Rosa26-eYFP mice (Figure 1A,B). (P^3^)PP treatment had no obvious effect on body weight in either mouse model (Figure 1 and Figure 2). Cumulative food intake was decreased (*p* < 0.05) in STZ-diabetic Ins*^Cre/+^*;Rosa26-eYFP mice (Figure 1C), but not in Glu*^CreERT2^*;Rosa26-eYFP mice (Figure 2C). Notably, (P^3^)PP intervention restored food intake to control levels in Ins1*^Cre/+^*;Rosa26-eYFP mice (Figure 1C). As expected, STZ increased (*p* < 0.05–0.001) blood glucose levels and reduced (*p* < 0.05–0.01) pancreatic insulin content in both transgenic mouse models (Figure 1D,E and Figure 2D,E). (P^3^)PP did not affect circulating glucose concentrations, but did significantly elevate (*p* < 0.05) pancreatic insulin content in both models (Figure 1D,E and Figure 2D,E).

### 2.2. Effects of (P^3^)PP on Islet Morphology in STZ-Diabetic Ins1^Cre/+^;Rosa26-eYFP and Glu^CreERT2^;Rosa26-eYFP Mice

The impact of multiple low-dose STZ on islet architecture was similar in Ins1*^Cre/+^*;Rosa26-eYFP and Glu*^CreERT2^*;Rosa26-eYFP mice (Figure 3 and Figure 4). As such, STZ reduced (*p* < 0.05–0.001) islet and beta-cell areas, with subsequent elevation (*p* < 0.05–0.001) of alpha-cell area and the alpha:beta cell ratio (Figure 3 and Figure 4). Representative images for pancreatic islets stained for insulin and glucagon are shown in panel E of Figure 3 and Figure 4. (P^3^)PP intervention exerted equivalent positive effects on islet morphology in both mouse models. Specifically, islet area was increased by 11-day twice-daily (P^3^)PP administration, significantly so (*p* < 0.05) in Glu*^CreERT2^*;Rosa26-eYFP mice (Figure 3A and Figure 4A), with a corresponding enhancement (*p* < 0.05–0.01) of beta-cell area (Figure 3B and Figure 4B). (P^3^)PP also decreased (*p* < 0.05) STZ-induced elevations of alpha-cell area in both Ins1*^Cre/+^*;Rosa26-eYFP and Glu*^CreERT2^*;Rosa26-eYFP mice (Figure 3C and Figure 4C), that together resulted in a near normalisation of the alpha:beta cell ratio in these mice (Figure 3D and Figure 4D).

### 2.3. Effects (P^3^)PP on Islet Cell Turnover and Plasticity in STZ-Diabetic Ins1^Cre/+^;Rosa26-eYFP and Glu^CreERT2^;Rosa26-eYFP Mice

Beta-cell proliferation was decreased (*p* < 0.001) by STZ in Ins1*^Cre/+^*;Rosa26-eYFP mice (Figure 5A), but not in Glu*^CreERT2^*;Rosa26-eYFP mice (Figure 6A). However, as would be expected, STZ-induced beta-cell apoptosis rates were increased (*p* < 0.001) in the transgenic mouse models (Figure 5B and Figure 6B). (P^3^)PP restored beta-cell turnover rates similar to those observed in normal healthy mice in both STZ-diabetic Ins1*^Cre/+^*;Rosa26-eYFP and Glu*^CreERT2^*;Rosa26-eYFP mice (Figure 5A,B and Figure 6A,B). Alpha-cell proliferation and apoptosis were increased (*p* < 0.01–0.001) by STZ in both models, with (P^3^)PP administration returning alpha-cell turnover rates towards control levels (Figure 5C,D and Figure 6C,D). Interestingly, in Ins1*^Cre/+^*;Rosa26-eYFP mice, beta-cell de-differentiation, as well as the transdifferentiation of beta-cells toward an alpha-cell phenotype, was augmented (*p* < 0.05–0.001) by STZ (Figure 5E,F). These detrimental STZ-induced changes on islet cell lineage were almost fully reversed by (P^3^)PP treatment (Figure 5E,F). In Glu*^CreERT2^*;Rosa26-eYFP mice, STZ increased (*p* < 0.01) alpha-cell de-differentiation, as well promoting the transdifferentiation of alpha-cells to beta-cells (Figure 6E,F), with these changes in islet cell plasticity being further augmented (*p* < 0.05) by twice-daily (P^3^)PP treatment (Figure 6E,F).

## 3. Discussion

The primary objective of the current study was to further probe the islet morphology benefits of sustained NPY4R modulation [2,4], utilising (P^3^)PP as a long-acting NPY4R agonist [3]. We employed transgenic Ins1*^Cre/+^*;Rosa26-eYFP and Glu*^CreERT2^*;Rosa26-eYFP mouse models that permit for lineage tracing of islet beta- and alpha-cells, respectively [5,12]. Thus, both the onset of diabetes, as well as the mechanism of action of several experimental and clinically approved antidiabetic drugs, are thought to be linked to changes in islet cell plasticity [15,16,17]. In addition, there is a belief that NPY4R modulation can positively alter islet cell identity to help retain functional beta-cell mass [8,9,11].

As expected, multiple-low-dose STZ administration elevated blood glucose levels in both Ins1*^Cre/+^*;Rosa26-eYFP and Glu*^CreERT2^*;Rosa26-eYFP mice, as a result of specific pancreatic beta-cell destruction, resulting in severely depleted pancreatic insulin stores [18], with this insulin deficiency also leading to anticipated weight loss. Small differences were apparent between the transgenic mouse models in terms of STZ impact on body weight and food intake, suggesting marginally more detrimental effects in Ins1*^Cre/+^*;Rosa26-eYFP mice, as has been observed previously [19]. Indeed, the consistency of STZ actions in the current setting shows the reliability and transparency of our findings. Moreover, given that STZ acts as a highly toxic DNA alkylating agent [18], it would seem unlikely that presence of the transgene in Ins1*^Cre/+^*;Rosa26-eYFP mice affects their susceptibility to the toxin. Indeed, studies in Glu*^CreERT2^*;ROSA26-eYFP mice confirm that activation of the Cre recombinase enzyme has no impact on normal islet cell processes [20]. Despite this, we are unable to rule out the possibility, although improbable, that GFP production within beta-cells of Ins1*^Cre/+^*;Rosa26-eYFP mice could increase metabolic demand, leaving these cells more susceptible to STZ. In addition, Glu*^CreERT2^*;Rosa26-eYFP mice also require a single, low-dose, tamoxifen injection to induce Cre-lox recombination, but since oestrogen has well-documented benefits on islet cell function [21], any potential effect of tamoxifen to protect Glu*^CreERT2^*;*Rosa26-eYFP* mice from the detrimental effects of STZ is questionable. Thus, a more plausible explanation for the slight variation in STZ sensitivity between the two transgenic mouse models appears to relate to previously documented differences between [22], and even within [23], strains of mice administered low-dose STZ due to differences in genetic background.

Furthermore, whilst utilising two separate transgenic mouse models is a major strength of the current study, permitting for lineage tracing of both islet alpha- and beta-cells, it is important to note that these models are not exact replicas of one another. Specifically, Ins1*^Cre/+^*;Rosa26-eYFP mice have a constitutively active beta-cell reporter gene, whereas Glu*^CreERT2^*;Rosa26-eYFP mice possess an inducible reporter gene in their alpha-cells. Thus, as noted previously [19], some care may be advisable when making direct comparisons between both models with initial body weights also showing some slight variation, but this in no way detracts from clear and noteworthy observations made within each transgenic model. It is interesting that (P^3^)PP intervention in Ins1*^Cre/+^*;Rosa26-eYFP mice reversed STZ-induced reductions in calorie intake, despite being principally viewed as an anorectic hormone [24]. This likely reflects the severity of the multiple-low-dose STZ model of diabetes [18], and the propensity for NPY4R to restore normal metabolism. Either way, both transgenic mouse models displayed an equivalent impact of STZ at the level of the pancreatic islet, which included changes to islet cell plasticity, validating their use to undertake a more comprehensive investigation of the impact of sustained NPY4R activation on islet cell identity and overall morphology. As an alternative approach, high-fat feeding could have been employed in the Ins1*^Cre/+^*;Rosa26-eYFP and Glu*^CreERT2^*;Rosa26-eYFP transgenic mice to induce islet cell stress and lineage alterations [17]. However, this type of dietary intervention tends not to lead to hyperglycaemia, and islet stress occurs over a more prolonged time period, making it difficult to pinpoint the exact timing of key islet cell transitioning events.

Earlier studies with (P^3^)PP in a rodent model of obesity-diabetes demonstrated treatment-induced preservation of pancreatic islet morphology and enhanced insulin secretion [4]. These positive effects on islet architecture were linked to increased beta-cell proliferation and decreased apoptosis [4], but it seems likely that alteration of islet cell plasticity would also be important in this regard. Moreover, sustained NPY1R activation has previously been demonstrated to drive pancreatic alpha- to beta-cell transdifferentiation in STZ-diabetic mice, in a bid to restore beta-cell mass [19]. Although clinical application of NPY1R-specific compounds for diabetes is limited by off-target side-effects including increased appetite [25] and induction of hypertension [26], which is not the case for NPY4R, the fact that the intracellular signalling cascades triggered following NPY1R or NPY4R activation are comparable [27] supports the likely positive effects of (P^3^)PP on islet cell lineage.

In good accord with our initial observations [4], 11-day twice-daily (P^3^)PP treatment partially reversed the negative impact of STZ on pancreatic islet architecture in both STZ-diabetic Ins1*^Cre/+^*;Rosa26-eYFP and Glu*^CreERT2^*;Rosa26-eYFP mice, leading to a significant increase in pancreatic insulin content. It is noteworthy, however, that the relative enhancement of pancreatic insulin content in both mouse models was comparatively modest when compared to changes in islet morphology, suggesting investigation of beta-cell health through assessment of key transcription factors such as Pdx1 or Mafa might be informative [28]. In addition, although beta-cell area was enhanced by (P^3^)PP treatment in both mouse models, STZ-induced impairment of the classic murine islet ‘alpha-cell mantle/beta-cell core’ was still evident. That said, the clear augmentation of beta-cell area was consistently associated with almost complete protection against apoptosis, as well as positive effects on beta-cell proliferation in Ins1*^Cre/+^*;Rosa26-eYFP mice. In line with the previously documented glucagonostatic actions of PP [29,30], (P^3^)PP treatment curbed STZ-induced elevations of pancreatic alpha-cell area, an observation that also fits well with earlier studies [4], this being despite prominent reductions in alpha-cell apoptotic rates in (P^3^)PP-treated STZ-diabetic transgenic mice, once more implying that modulation of islet cell lineage is more relevant for (P^3^)PP-induced alterations of islet morphology.

Thus, in terms of islet cell lineage tracing, beta-cell de-differentiation and the transdifferentiation of beta-cells towards an alpha-cell like phenotype was promoted by STZ administration, in good agreement with previous work [17,31]. Notably, (P^3^)PP administration was able to restrain these changes in beta-cell plasticity, encouraging such cells to persist as mature insulin-secreting beta-cells, despite STZ insult. Accordingly, in Glu*^CreERT2^*;Rosa26-eYFP mice, treatment with (P^3^)PP was confirmed to promote utilisation of the pancreatic alpha-cell pool for beta-cell regeneration. This principle fits well with (P^3^)PP-mediated reductions in STZ-mediated alpha-cell expansion. It would be intriguing to assess modifications of PP expression in relation to islet cell plasticity changes using our immunohistochemical approaches, but such investigations are hindered by use of long-acting (P^3^)PP that may be internalised after receptor binding and that exhibits extremely close amino acid sequence homology to native PP. Interestingly, induction of diabetes per se also led to augmented alpha-cell de-differentiation, as well as transdifferentiation to insulin-positive beta-cells, likely as an inherent adaptive response to STZ induced beta-cell ablation [5]. The possibility of beta-cell neogenesis from non-islet cells was not explored in the current transgenic mouse models, but previous work indicates that the contribution of duct cells towards normal beta-cells and functional islets is minimal [32]. Importantly, given harvest of pancreatic tissues occurred at a point where improvements in glycaemic status were not apparent, it is reasonable to assume direct NPY4R-mediated effects of (P^3^)PP on islet cell plasticity, since correction of hyperglycaemia has been shown to independently affect islet cell lineage [33]. Taken together, the positive impact of (P^3^)PP on islet cell lineage is reminiscent of that yielded by NPY1R activation [19], in keeping with stimulation of complementary cell signalling pathways [34,35,36], and positions (P^3^)PP alongside other established antidiabetic drugs, such as the incretin enhancers, known to preserve mature islet beta-cell identity [16,17]. Overall, PP has not been extensively studied to date, with limited knowledge on mechanisms and related effects [36,37]; the current study helps to address this issue and confirms important NPY4R-mediated actions on energy regulation and pancreatic endocrine function, with important effects on islet cell lineage that support conservation of normal pancreatic islet morphology.

## 4. Materials and Methods

### 4.1. Peptides

(P^3^)PP, an enzymatically stable PP analogue with substitution of Leu3 for Pro3 [3], was obtained from Synpeptide Co., Ltd. (Shanghai, China) at 95% purity. Peptide purity and identity was confirmed in-house by high-performance liquid chromatography (HPLC) and Matrix-Assisted Laser Desorption/Ionisation Time-of-Flight Mass Spectrometry (MALDI–ToF MS), respectively, as described previously [3].

### 4.2. Animals

Ins1*^Cre/+^*;Rosa26-eYFP and Glu*^CreERT2^*;Rosa26-eYFP transgenic mice are fully characterised by [20,38], respectively, and subsequently effectively employed within our laboratory [5,12]. All experiments were conducted under the UK Animals (Scientific Procedures) Act 1986 and EU Directive 2010/63EU, as well as being approved by the local Ulster University Animal Welfare and Ethical Review Body (AWERB). Animals were used at 12–14 weeks of age and were maintained in an environmentally controlled unit at 22 ± 2 °C with a 12 h dark and light cycle and given ad libitum access to a standard rodent diet (10% fat, 30% protein and 60% carbohydrate; Trouw Nutrition, Northwich, UK) and drinking water. Where appropriate, beta-cell destruction and insulin deficiency was induced by low-dose STZ injection (50 mg/kg body weight, i.p.) for 5 consecutive days, with (P^3^)PP treatment intervention occurring 2 days prior to STZ administration. Glu*^CreERT2^*;Rosa26-eYFP mice were also administered tamoxifen (7 mg/mouse bw, i.p.) 5 days prior to STZ intervention to induce expression of the alpha-cell fluorescent lineage marker protein, whereas enhanced yellow fluorescent protein (eYFP) is constitutively active within the beta-cells of Ins*^Cre/+^*;Rosa26-eYFP mice. Following tamoxifen administration, around 70% of glucagon-positive alpha-cells have detectable green fluorescent protein (GFP) expression in STZ-diabetic Glu*^CreERT2^*;Rosa26-eYFP mice [12], with the *Cre* recombinase enzyme confirmed to be specific to only islet alpha-cells [39], substantiating the appropriateness of this model to study alpha-cell transition events. Male mice were employed for all studies given the sex-based impact of STZ on pancreatic islet cell adaptions, which importantly also includes differences in rates of cellular transdifferentiation events [40].

### 4.3. Experimental Protocols

Ins*^Cre/+^*;Rosa26-eYFP and Glu*^CreERT2^*;Rosa26-eYFP male mice (*n* = 7) received twice-daily (09:00 and 17:00 h) treatment with either saline vehicle (0.9% (*w*/*v*) NaCl) or (P^3^)PP (25 nmol/kg bw) 2 days prior to the first STZ injection, and throughout the full duration of the study. This dosing regimen was based on previous studies with (P^3^)PP in a rodent model of obesity-diabetes [4]. For both Ins*^Cre/+^*;Rosa26-eYFP and Glu*^CreERT2^*;Rosa26-eYFP mice, body weight, cumulative food intake and blood glucose were assessed at regular intervals. At the end of the treatment period, animals were culled by cervical dislocation and pancreatic tissues were excised, divided longitudinally, and processed for either determination of pancreatic insulin content following acid/ethanol protein extraction or fixed in 4% paraformaldehyde for 48 h at 4 °C for histological analysis.

### 4.4. Immunohistochemistry

Fixed pancreatic tissues were embedded in paraffin wax blocks using an automated tissue processor (Leica TP1020, Leica Microsystems, Buffalo Grove, IL, USA); 5 μm sections were cut (Shandon Finesse 325 microtome, Thermo Scientific, Waltham, MA, USA) and mounted onto poly-L-lysine coated glass slides. Slides were dewaxed by immersion in xylene and rehydrated through a series of ethanol solutions of reducing concentration (100–50%). Heat-mediated antigen retrieval was then carried out in citrate buffer. Sections were blocked in 4% bovine serum albumin solution before 4 °C overnight incubation with appropriate primary antibodies including insulin (1:400; Abcam, ab6995, Cambridge, UK), glucagon (1:400; raised in-house, PCA2/4), GFP (1:1000; Abcam, ab5450) or Ki-67 (1:500; Abcam, ab15580), as described previously [19]. Islet cell apoptosis was determined using co-expression of terminal deoxynucleotidyl transferase dUTP nick end labelling (TUNEL) with either insulin or glucagon, whereas co-expression of Ki-67 with either insulin or glucagon was employed for cellular proliferation analysis. For cell lineage studies and assessment of islet cell de- and trans-differentiation, the proportion of alpha- or beta-cells positive for GFP was determined, where our GFP antibody (Abcam, ab5450) is reactive against all variants of *Aequorea victoria* GFP, including eYFP. Subsequently, slides were then rinsed in PBS and incubated for 45 min at 37 °C with appropriate Alexa Fluor secondary antibodies (1:400; Invitrogen (Waltham, MA, USA), Alexa Fluor 488 for green or 594 for red). Slides were finally incubated with DAPI for 15 min at 37 °C, and then mounted for imaging using a fluorescent microscope (Olympus model BX51, Tokyo, Japan) fitted with DAPI (350 nm), FITC (488 nm) and TRITC (594 nm) filters and a DP70 camera adapter system (Olympus, Tokyo, Japan).

### 4.5. Image Analysis

Analysis of islet parameters, including islet area, beta cell area, alpha cell area (expressed in μm^2^), and the ratio of alpha to beta cells, was conducted using Cell^F^ imaging software version 1.17 (Olympus, Tokyo, Japan). For quantification of apoptosis in beta and alpha cells, the number of cells positive for insulin or glucagon and TUNEL was counted. A similar approach was used for analysing cell proliferation, where Ki-67 was co-stained with insulin or glucagon. For appraisal of changes in cell lineage, cells co-expressing both insulin and GFP (insulin^+ve^, GFP^+ve^ cells), cells expressing insulin with no GFP (insulin^+ve^, GFP^−ve^ cells), cells expressing glucagon without GFP (glucagon^−ve^, GFP^+ve^ cells), and cells co-expressing GFP and glucagon (glucagon^+ve^, GFP^+ve^ cells) were analysed, as appropriate. As such, cells positive for both glucagon and GFP in Ins1*^Cre/+^*;Rosa26-eYFP mice were considered to have transdifferentiated from a beta-cell to an alpha-cell phenotype, whereas positive co-staining for insulin and GFP in Glu*^CreERT2^*;Rosa26-eYFP mice represented transdifferentiation of an alpha-cell to a beta-cell. All cell counts were conducted in a blinded manner, with more than 50 islets analysed per treatment group and morphometric analysis performed on every tenth section to ensure different islets were analysed.

### 4.6. Biochemical Analysis

Blood samples were collected from the cut tail vein of animals. Blood glucose was measured using a portable Ascencia Contour blood glucose meter (Bayer Healthcare, Newbury, Berkshire, UK). For pancreatic insulin content, snap frozen pancreatic tissues were homogenised in acid/ethanol (75% (*v*/*v*) ethanol, distilled water and 1.5% (*v*/*v*) 12 M HCl) and protein was extracted in a pH neutral Tris (hydroxymethyl) aminomethane (TRIS) buffer, with protein content determined using Bradford reagent (Sigma-Aldrich, St. Louis, MO, USA) and insulin content determined by an in-house insulin radioimmunoassay [3].

### 4.7. Statistics

Data analysis was carried out on GraphPad PRISM software (version 8.0), with results presented as mean ± standard error of the mean (SEM). Statistical comparisons between experimental groups were conducted using a one-way or two-way ANOVA, as appropriate, utilising Bonferroni post hoc tests for multiple comparisons between treatment groups. A *p*-value of less than 0.05 was considered statistically significant, with the asterisk symbols on all figures denoting a significant difference when compared to the respective STZ-diabetic control group of mice.

## 5. Conclusions

The present study has established that pancreatic islet benefits of prolonged (P^3^)PP administration are linked to reduced beta-cell apoptosis and increased proliferation in STZ-diabetic mice. Furthermore, this is the first study to reveal a direct positive influence of NPY4R activation on islet cell transitioning events. We have confirmed that NPY4R agonists impart disease-modifying advantages on pancreatic islets in diabetes, associated with positive actions on the plasticity of both alpha- and beta-cells.

## Figures and Tables

**Figure 1 ijms-26-04215-f001:**
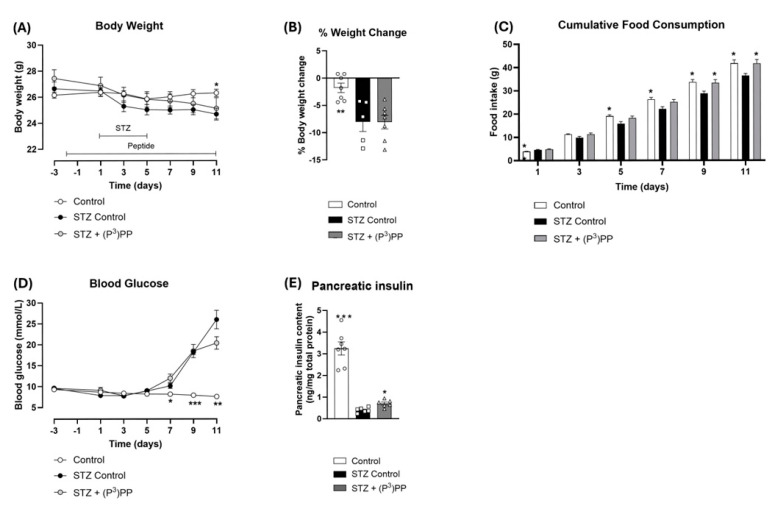
Effects of twice-daily (P^3^)PP administration on body weight, energy intake, circulating glucose and pancreatic insulin content in STZ-diabetic Ins1^Cre/+^;Rosa26-eYFP mice. Body weight (**A**), percentage body weight change (**B**), cumulative energy intake (**C**), circulating blood glucose (**D**) and pancreatic insulin content (**E**) were measured during (**A**,**C**,**E**) or after (**B**,**D**) twice-daily treatment with saline vehicle (0.9% NaCl) or (P^3^)PP (25 nmol/kg bw, i.p.) for 11 days in STZ-diabetic Ins1^Cre/+^;Rosa26-eYFP transgenic mice. Values are mean ± SEM for n = 7 mice. * *p* < 0.05, ** *p* < 0.01 and *** *p* < 0.001 compared with STZ-diabetic controls. (**A**,**C**,**D**) Datasets were analysed using a two-way repeated-measures ANOVA with a Bonferroni post hoc test or (**B**,**E**) using a one-way ANOVA with a Bonferroni post hoc test.

**Figure 2 ijms-26-04215-f002:**
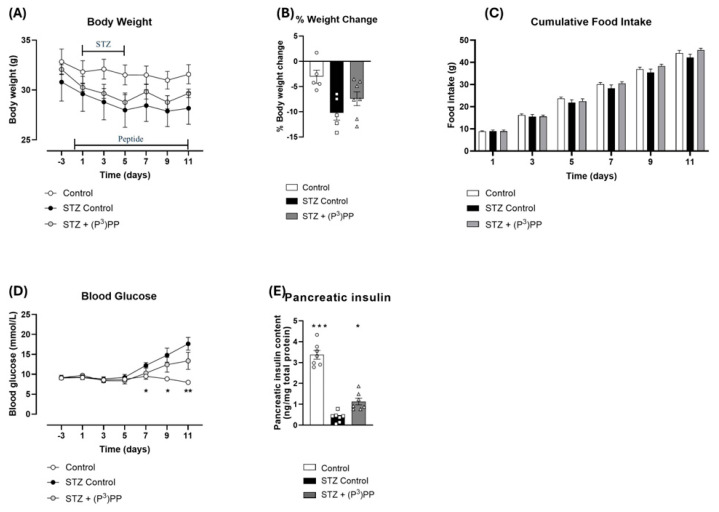
Effects of twice-daily (P^3^)PP administration on body weight, energy intake, circulating glucose and pancreatic insulin content in STZ-diabetic Glu*^CreERT2^*;Rosa26-eYFP mice. Body weight (**A**), percentage body weight change (**B**), cumulative energy intake, (**C**) circulating blood glucose (**D**) and pancreatic insulin content (**E**) were measured during (**A**,**C**,**E**) or after (**B**,**D**) twice-daily treatment with saline vehicle (0.9% NaCl) or (P^3^)PP (25 nmol/kg bw, i.p.) for 11 days in STZ-diabetic Glu*^CreERT2^*;Rosa26-eYFP transgenic mice. Values are mean ± SEM for *n* = 7 mice. * *p* < 0.05, ** *p* < 0.01 and *** *p* < 0.001 compared with STZ-diabetic controls. (**A**,**C**,**D**) Datasets were analysed using a two-way ANOVA with Bonferroni post hoc test or (**B**,**E**) using a one-way, repeated-measures ANOVA with a Bonferroni post hoc test.

**Figure 3 ijms-26-04215-f003:**
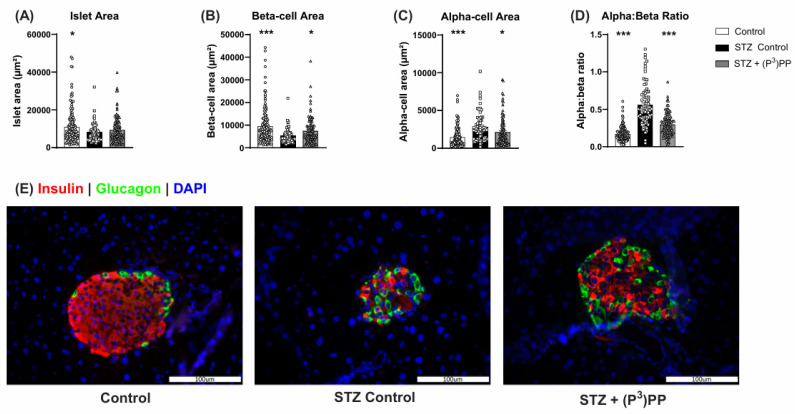
Effects of twice-daily (P^3^)PP administration on pancreatic islet morphology in STZ-induced diabetic Ins1*^Cre/+^*;Rosa26-eYFP mice. Islet (**A**), beta-cell (**B**) and alpha-cell (**C**) areas as well as alpha:beta ratio (**D**) were measured using Cell^F^ image analysis software version 1.17 (Olympus, Tokyo, Japan) after 11 days of twice-daily treatment with saline vehicle (0.9% NaCl) or (P^3^)PP (25 nmol/kg bw, i.p.) in STZ-diabetic Ins1*^Cre/+^*;Rosa26-eYFP transgenic mice. (**E**) Representative images (40×) of islets showing insulin (red), glucagon (green) and DAPI (blue) immunoreactivity from each group of mice. Values are mean ± SEM for 7 mice. * *p* < 0.05 and *** *p* < 0.001 compared with STZ-diabetic controls. Datasets were analysed using a one-way ANOVA with a Bonferroni post hoc test.

**Figure 4 ijms-26-04215-f004:**
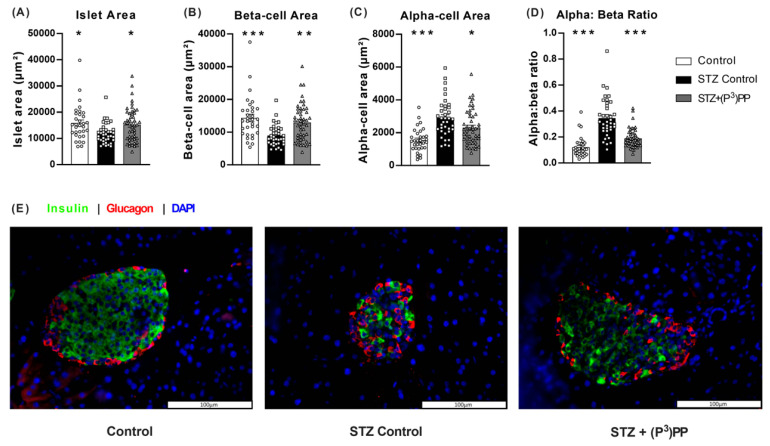
Effects of twice-daily (P^3^)PP administration on pancreatic islet morphology in STZ-induced diabetic Glu*^CreERT2^*;Rosa26-eYFP mice. Islet (**A**), beta-cell (**B**) and alpha-cell (**C**) areas as well as alpha:beta ratio (**D**) were measured using Cell^F^ image analysis software after 11-day twice-daily treatment with saline vehicle (0.9% NaCl) or (P^3^)PP (25 nmol/kg bw, i.p.) in STZ-diabetic Glu*^CreERT2^*;Rosa26-eYFP transgenic mice. (**E**) Representative images (40×) of islets showing insulin (red), glucagon (green) and DAPI (blue) immunoreactivity from each group of mice. Values are mean ± SEM for 7 mice. * *p* < 0.05, ** *p* < 0.01 and *** *p* < 0.001 compared with STZ-diabetic controls. Datasets were analysed using a one-way ANOVA with a Bonferroni post hoc test.

**Figure 5 ijms-26-04215-f005:**
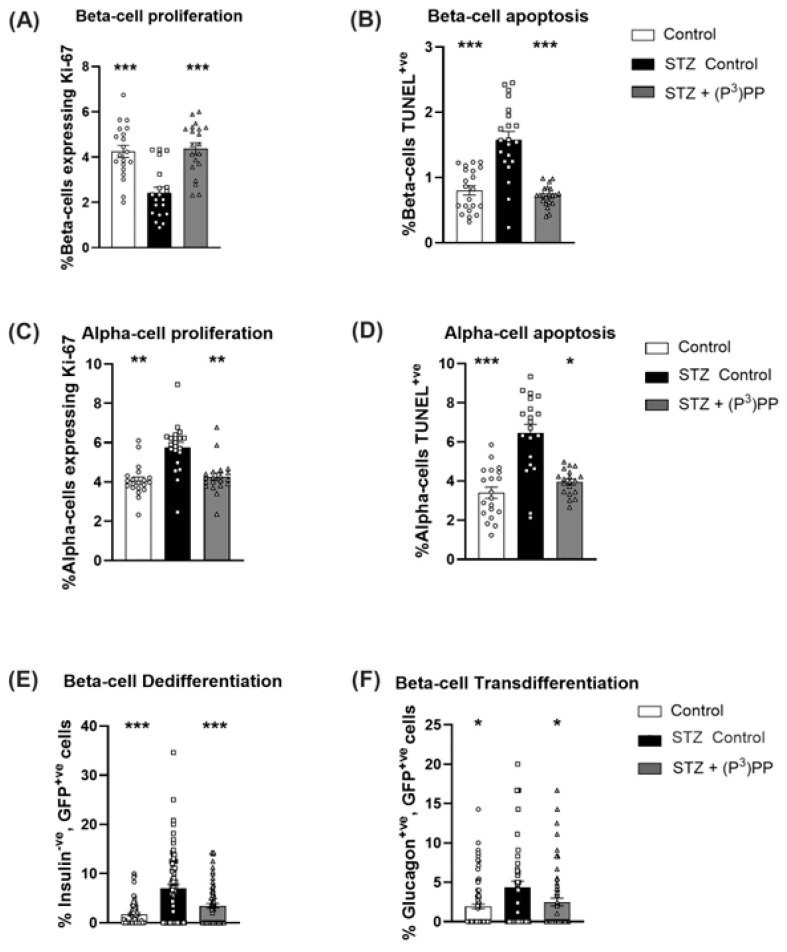
Effects of twice-daily (P^3^)PP administration on islet cell turnover and plasticity in STZ-induced diabetic Ins1*^Cre/+^*;Rosa26-eYFP mice. Beta-cell proliferation (**A**) and apoptosis (**B**) as well as alpha-cell proliferation (**C**) and apoptosis (**D**) alongside beta-cell dedifferentiation (**E**) and transdifferentiation (**F**) were measured after 11-day twice-daily treatment with saline vehicle (0.9% NaCl) or (P^3^)PP (25 nmol/kg bw, i.p.) in STZ-diabetic Ins1*^Cre/+^*;Rosa26-eYFP transgenic mice. Cell proliferation (**A**,**C**) was assessed using Ki-67 staining, with apoptosis rates (**B**,**D**) analysed by TUNEL staining. Beta-cell dedifferentiation (**E**) was assessed by counting islet cells positive for GFP but negative for insulin, with beta-cell transdifferentiation (**F**) measured by counting islet cells co-positive for GFP and glucagon. Values are mean ± SEM for 7 mice. * *p* < 0.05, ** *p* < 0.01 and *** *p* < 0.001 compared with STZ-diabetic controls. Datasets were analysed using a one-way ANOVA with a Bonferroni post hoc test.

**Figure 6 ijms-26-04215-f006:**
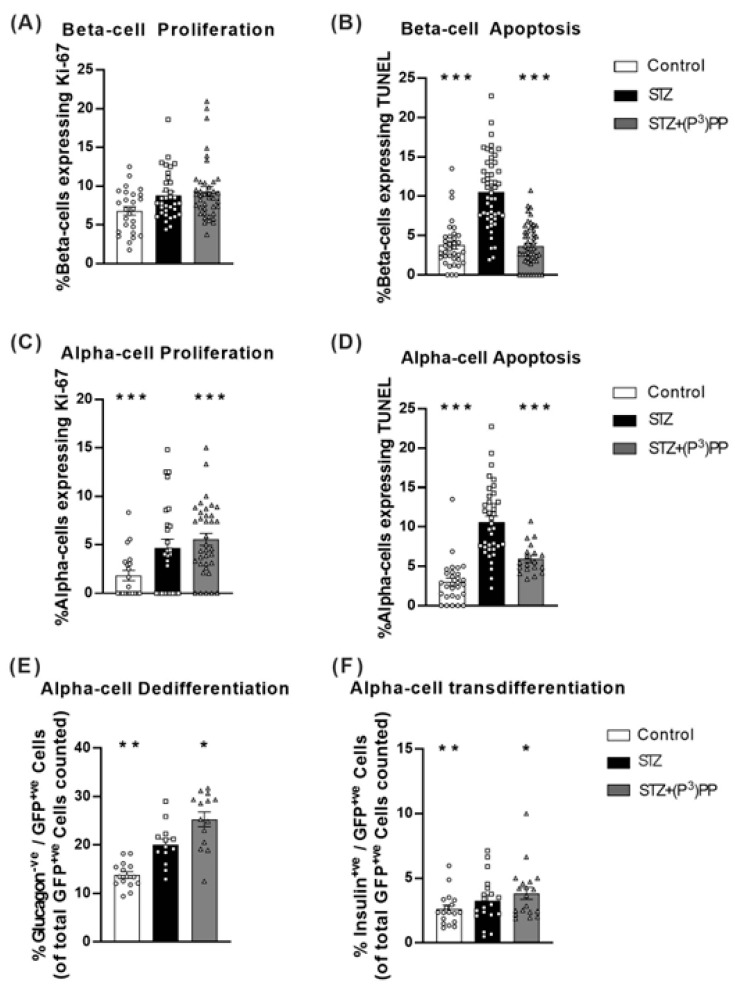
Effects of twice-daily (P^3^)PP administration on islet cell turnover and plasticity in STZ-induced diabetic Glu*^CreERT2^*;Rosa26-eYFP mice. Beta-cell proliferation (**A**) and apoptosis (**B**) as well as alpha-cell proliferation (**C**) and apoptosis (**D**) alongside alpha-cell dedifferentiation (**E**) and transdifferentiation (**F**) were measured after 11-day twice-daily treatment with saline vehicle (0.9% NaCl) or (P^3^)PP (25 nmol/kg bw, i.p.) in STZ-diabetic Glu*^CreERT2^*;Rosa26-eYFP transgenic mice. Cell proliferation (**A**,**C**) was assessed using Ki-67 staining with apoptosis rates (**B**,**D**) analysed by TUNEL staining. Alpha-cell dedifferentiation (**E**) was assessed by counting islet cells positive for GFP but negative for glucagon, with alpha-cell transdifferentiation (**F**) measured by counting islet cells co-positive for GFP and insulin. Values are mean ± SEM for 7 mice. * *p* < 0.05, ** *p* < 0.01 and *** *p* < 0.001 compared with STZ-diabetic controls. Datasets were analysed using a one-way ANOVA with a Bonferroni post hoc test.

## Data Availability

The authors declare that the data supporting the findings of this study are available within the article. Any additional raw data supporting the conclusions of this article will be made available by the lead author, without undue reservation.

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
