# Peer review of "The Beneficial Impact of a Novel Pancreatic Polypeptide Analogue on Islet Cell Lineage"

_ijms, 2025, doi:10.3390/ijms26094215_

Round 1

Reviewer 1 Report

Comments and Suggestions for Authors

In this paper, the authors investigated the molecular effects of a novel NPY4R-specific PP analogue on islet cells using cell lineage tracing in mice. They found the analogue had a tendency of reversing the impaired islet cell lineages at least partially in low-dose STZ mice models. PP is still a peptide of mystery, and such a study is quite interesting. However, I have several concerns,

1) There is an apparent discrepancy of phenotypes of two transgenic mouse lines. The authors said this had been already observed in a previous paper, but another discussion in this paper is needed. If genetic backgrounds of the two lines are identical, one possibility is that a high-level expression of the large YFP protein in the beta or alpha cells interfered the normal function in any way, since the insulin on glucagon promoter activity is supposed to be strong.

2)Since PP is expressed in gamma cells in the islets, it is an interesting issue whether such PP actions also work as an intrinsic protective pathway. In this regard, the expression of PP in the islets in these models may be important. More specifically, the identity of de-differentiated beta or alpha cells is of great interest, and whether they are PP-expressing precursor cells which may serve as the restoration of beta cells after injury is a significant issue. Did the authors have such results?

3)As for the islet cell lineage, we would like to know whether islet or beta cell neogenesis from non-islet cells such as pancreatic ducts occurred in low-dose STZ models with or without PP treatment.

3)In Figure 1, the insulin area of the low-dose STZ was recovered to control levels after PP treatment, but the insulin content did not. We would like to know what is lacking in this partial recovery, such as reduced Pdx1 or Mafa levels. We would also like to know low-dose STZ caused irreversible damage to the beta cells, which cannot be overcome by PP treatment.

5)Figure 2 and 5: The islet size seemed smaller in low-dose STZ group than the others, but was this the case? Please describe the size distribution of islets in order not to mislead the readers.

Author Response

Referee 1

Comment: In this paper, the authors investigated the molecular effects of a novel NPY4R-specific PP analogue on islet cells using cell lineage tracing in mice. They found the analogue had a tendency of reversing the impaired islet cell lineages at least partially in low-dose STZ mice models. PP is still a peptide of mystery, and such a study is quite interesting.

Author reply: We agree that progressing knowledge around PP is a worthy study objective, and that the Referee finds our data interesting.

Comment: However, I have several concerns, 1) There is an apparent discrepancy of phenotypes of two transgenic mouse lines. The authors said this had been already observed in a previous paper, but another discussion in this paper is needed. If genetic backgrounds of the two lines are identical, one possibility is that a high-level expression of the large YFP protein in the beta or alpha cells interfered the normal function in any way, since the insulin on glucagon promoter activity is supposed to be strong.

Author reply: The authors thank the Reviewer for this comment and are happy to comply with his/her suggestion. We agree that a more severe diabetic phenotype was apparent in Ins1Cre/+;Rosa26-eYFP transgenic mice as compared to GluCreERT2;Rosa26-eYFP mice. Naturally we would have preferred the phenotypes to be perfectly matched. But on the positive side, we believe that the slight difference in response to STZ allows us to dissociate between direct benefits of our test compound, (P3)PP, on islet cell lineage, against changes that are dependent on improved glycaemic control. Also, we considered it valuable to include both arms of study in a single manuscript, with appropriate discussion, rather than producing two separate papers. Whilst we think it unlikely that the YFP transgene protein could interfere with the actions of STZ, we have included some commentary in this regard. To highlight this point in the revised manuscript and directly address the comment raised by the Reviewer, the following sentences have been added to the Discussion section, that read (Page 12, lines 20-22 and Page 13 lines 1-12): ‘Moreover, given that STZ acts as a highly toxic DNA alkylating agent [23], it would seem unlikely that presence of the transgene in Ins1Cre/+;Rosa26-eYFP mice affects their susceptibility to the toxin. Indeed, studies in GluCreERT2;ROSA26-eYFP mice confirm that activation of the Cre recombinase enzyme has no impact on normal islet cell processes [16]. Despite this, we are unable to rule out the possibility, although improbable, that GFP protein production within beta-cells of Ins1Cre/+;Rosa26-eYFP mice could increase metabolic demand leaving these cells more susceptible to STZ. In addition, GluCreERT2;Rosa26-eYFP mice also require a single, low-dose, tamoxifen injection to induce Cre-lox recombination, but since oestrogen has well documented benefits on islet cell function [24], any potential effect of tamoxifen to protect GluCreERT2;Rosa26-eYFP mice from the detrimental effects of STZ is questionable. Thus, a more plausible explanation for the slight variation of STZ sensitivity between the two transgenic mouse models appears to relate to previously documented differences between [25], and even within [26], strains of mice administered low-dose STZ due to differences in genetic background’. The reference section has been updated accordingly.

Comment: 2) Since PP is expressed in gamma cells in the islets, it is an interesting issue whether such PP actions also work as an intrinsic protective pathway. In this regard, the expression of PP in the islets in these models may be important. More specifically, the identity of de-differentiated beta or alpha cells is of great interest, and whether they are PP-expressing precursor cells which may serve as the restoration of beta cells after injury is a significant issue. Did the authors have such results?

Author reply: The authors thank the Reviewer for this comment, and agree that investigating whether de-differentiated beta- and alpha-cells express PP would be of interest, as well as assessing islet PP expression in general. However, our treatment modality, namely the long-acting PP analogue (P3)PP, directly hinders such analyses given its high sequence homology with the native PP sequence and likelihood for antibody cross-reactivity. Even so, to address the interesting point raised by the Reviewer, the authors have added the following sentence to the Discussion section of the revised manuscript, that reads (Page 16, lines 5-9): ‘It would be intriguing to assess modifications of PP expression in relation to islet cell plasticity changes using our immunohistochemical approaches, but such investigations are hindered by use of long-acting (P3)PP that may be internalised after receptor binding and exhibits extremely close amino acid sequence homology to native PP’.

Comment: 3) As for the islet cell lineage, we would like to know whether islet or beta cell neogenesis from non-islet cells such as pancreatic ducts occurred in low-dose STZ models with or without PP treatment.

Author reply: We thank the Reviewer for this comment. From our perspective, it has been recently noted that the contribution of duct cells towards normal beta-cells and functional islets is minimal (please see Liu et al. Experimental & Molecular Medicine, 53(4), 605–614; PMID: 33820959 for details). In addition, the primary reason for employing our transgenic mouse models was to assess changes in islet cell transition events, rather than beta-cell neogenesis from non-islet cells. That said, to highlight the point raised by the Reviewer within the manuscript, we are happy to add the following sentence to the Discussion section, that reads (Page 16, lines 12-14): ‘The possibility of beta-cell neogenesis from non-islet cells was not explored in the current transgenic mouse models, but previous work indicates that the contribution of duct cells towards normal beta-cells and functional islets is minimal [33]’. The reference section has been updated accordingly.

Comment: 4) In Figure 1, the insulin area of the low-dose STZ was recovered to control levels after PP treatment, but the insulin content did not. We would like to know what is lacking in this partial recovery, such as reduced Pdx1 or Mafa levels. We would also like to know low-dose STZ caused irreversible damage to the beta cells, which cannot be overcome by PP treatment.

Author reply: The Reviewer raises an interesting point, and we agree that islet and beta-cell areas recovered to close to normal levels in (P3)PP treated STZ-diabetic mice, but that total insulin content was still reduced. To appropriately determine levels of beta-cell transcription factors such as Pdx1 or Mafa, whilst avoiding any possible mis-interpretation, we would first need to sort these cells into their respective populations including original beta-cells, new beta-cells originating from an endocrine source as well as beta-cells that may have been generated from non endocrine duct cells. We do not have access to the necessary equipment to accurately sort such beta-cell populations, and in addition, we would need to repeat the entire study to investigate this aspect. Nevertheless, we do agree that further comment on the slight disparity between beta-cell area and pancreatic insulin content is warranted. The authors have therefore added the following sentence to the Discussion section of the revised manuscript, that reads (Page 15, lines 6-10): ‘It is noteworthy however, that the relative enhancement of pancreatic insulin content in both mouse models was comparatively modest when compared to changes in islet morphology, suggesting investigation of beta-cell health through assessment of key transcription factors such as Pdx1 or Mafa might be informative [31]’. The reference section has been updated accordingly.

In terms of beta-cell damage induced by STZ and how (P3)PP may affect this, we have not investigated these processes within the current study. The reason for employing the two transgenic mouse models in the current setting was to comprehensively understand effects of (P3)PP on islet cell lineage and related cellular plasticity, rather than beta-cell health directly. To avoid any potential misunderstanding, the authors have decided not to comment further on this matter within the revised manuscript. We trust the Reviewer will understand our reasoning for this.  

Comment: 5) Figure 2 and 5: The islet size seemed smaller in low-dose STZ group than the others, but was this the case? Please describe the size distribution of islets in order not to mislead the readers.

Author reply: The authors are a little unsure as to what the Reviewer is specifically referring to with this comment. Islet size was significantly decreased in all low-dose STZ groups of mice when compared to respective healthy controls. These data are clearly visible within panels A of Figures 2&5, and we specifically choose to display such graphs as scatter plots so that the Reader could easily see all datapoints and avoid any potential ambiguity. We subsequently selected representative islet images (panel E, Figures 2&5) to make it apparent to the Reader that low-dose STZ intervention reduced islet area. Importantly, we are confident that use of a scatterplots ensures that our graphs for islet, beta- and alpha-cell areas are not misleading.

Reviewer 2 Report

Comments and Suggestions for Authors

This study is based on the authors previous findings on [P3]PP and its agonist effect on the receptor NPY4R.
There are some points could be discussed for better transfering of the ideas from the authors to readers.

Two strains of transgenic mice were used, Ins1Cre/+;Rosa26-eYFP and GluCreERT2;Rosa26-eYFP, in this study. Suppose the islet cells from these two strains have eYFP signals (according to Ref. 17, 18 and published paper Diabetologia. 2017.  60(12): 2399), but in the manuscript, it seems it's not necessary to use these transgenic mice. For example, in Fig. 2 and Fig. 5, the alpha- and beta-cell identifications are based on insulin and glucagon immunostaining. Please explain the logic here in case I've missed any important information. If there're any data not shown or in supplementary information (www.mdpi.com/xxx/s1, nothing in here), which is not  supplied in the review panel, please let us know. In addition, in Fig. 6, it seems there's GFP strain of animal too, please give details about this strain in Methods and Materials.

Why there's big difference between the Fig. 1 and Fig. 4's initial bodyweight of the mice? Especially, the 2 control groups have about 5 grams difference averagely. This big difference induces uncertainties for those related parameters like food intake, FBG and many other subtle things (which is clear between Fig.1C/D and Fig.4C/D already), which could be affective factor in metabolism study. No matter the bodyweigh thing is due to age's difference or other reasons, that should be avoided.

From the authors, "... sustained NPY4R activation positively modulates beta-cell turnover, as well as islet cell plasticity", but it seems there's no direct evidence to announce beta-cell turnover and plasticity. More detailed information is need here.

For writing: 
The 3 in [P3]PP is differnt in text and in figures.
In Figure 4, the administration of Peptide and STZ should also be given.

Author Response

Referee 2

Comment: This study is based on the authors previous findings on (P3)PP and its agonist effect on the receptor NPY4R. There are some points could be discussed for better transfering of the ideas from the authors to readers. Two strains of transgenic mice were used, Ins1Cre/+;Rosa26-eYFP and GluCreERT2;Rosa26-eYFP, in this study. Suppose the islet cells from these two strains have eYFP signals (according to Ref. 17, 18 and published paper Diabetologia. 2017.  60(12): 2399), but in the manuscript, it seems it's not necessary to use these transgenic mice. For example, in Fig. 2 and Fig. 5, the alpha- and beta-cell identifications are based on insulin and glucagon immunostaining. Please explain the logic here in case I've missed any important information.

Author reply: We thank the Reviewer for this comment. He/she is correct that assessment of islet alpha- and beta-cell area does not require the transgene, being investigated using glucagon and insulin specific antibodies, respectively. However, presence of the transgene is necessary as a marker to study changes in islet cell lineage, which is the primary objective of the current study. Data pertaining to investigation of beta-cell lineage in Ins1Cre/+;Rosa26-eYFP mice are depicted within Figure 3, whilst data relating to changes in alpha-cell lineage using GluCreERT2;Rosa26-eYFP mice are displayed within Figure 6. We trust this explanation helps the Reviewer understand the need to the two different transgenic mouse models employed for the current study.

Comment: If there're any data not shown or in supplementary information (www.mdpi.com/xxx/s1, nothing in here), which is not  supplied in the review panel, please let us know.

Author reply: The authors can confirm that there is no supplementary data for this manuscript. All data are contained within the main manuscript.

Comment: In addition, in Fig. 6, it seems there's GFP strain of animal too, please give details about this strain in Methods and Materials.

Author reply: As noted in reply to the Reviewers first comment, there were two separate strains of mice used for the current study. Both these mouse models, namely Ins1Cre/+;Rosa26-eYFP and GluCreERT2;Rosa26-eYFP mice, are transgenic and contain a fluorescent transgene that is detectable using our GFP antibody. The fluorescent transgene is located within original beta-cells in Ins1Cre/+;Rosa26-eYFP mice, whereas the transgene is found in original alpha-cells in GluCreERT2;Rosa26-eYFP mice. Accordingly, cells staining for GFP mark beta-cells in the former, and alpha-cells in the latter, model.

Comment: Why there's big difference between the Fig. 1 and Fig. 4's initial bodyweight of the mice? Especially, the 2 control groups have about 5 grams difference averagely. This big difference induces uncertainties for those related parameters like food intake, FBG and many other subtle things (which is clear between Fig.1C/D and Fig.4C/D already), which could be affective factor in metabolism study. No matter the bodyweigh thing is due to age's difference or other reasons, that should be avoided.

Author reply: The authors thank the Reviewer for this comment, and agree that starting body weights between the two mouse models were slightly different. We can confirm that this variation is not due to different ages, and simply reflects inherent strain related differences. Importantly, as noted within the original manuscript, we clearly point out that the two transgenic models are not replicas of each other and that direct comparisons are best avoided. To add to this and directly address the comment raised by the Reviewer, we have now modified this section of text to read (Page 13, lines 17-21): ‘Thus, as noted previously [19], some care may be advisable when making direct comparisons between both models with initial body weights also showing some slight variation, but this in no way detracts from clear and noteworthy observations made within each transgenic model’.

Comment: From the authors, "... sustained NPY4R activation positively modulates beta-cell turnover, as well as islet cell plasticity", but it seems there's no direct evidence to announce beta-cell turnover and plasticity. More detailed information is need here.

Author reply: Apologies but the authors are confused by this comment. Our principal reason for employing the two chosen models, namely Ins1Cre/+;Rosa26-eYFP and GluCreERT2;Rosa26-eYFP transgenic mice, was to investigate effects of (P3)PP and NPY4R activation on islet cell plasticity – these data are contained with Figures 3&6, panels E&F. In terms of beta-cell turnover, we investigated effects of (P3)PP on beta-cell proliferation and apoptosis using Ki-67 and TUNEL stating, respectively, and these data are within panel A&B of Figure 3&6. The authors are content that our statement relating to the impact of sustained NPY4R activation on beta-cell turnover and islet cell plasticity is well founded.

Comment: For writing:  The 3 in (P3)PP is differnt in text and in figures.
Author reply: We thank the Reviewer for this comment and nomenclature for (P3)PP has now been standardised between the text and figures. The authors are thankful for this opportunity to improve the quality of our paper.

Comment: In Figure 4, the administration of Peptide and STZ should also be given.

Author reply: The authors are happy to add these details, in a similar fashion as depicted within Figure 1.

Reviewer 3 Report

Comments and Suggestions for Authors

In this work, the antidiabetic effect of a synthetic polypeptide none-digestible analogue (to pancreatic polypeptide, PP) synthetic polypeptide ([P3]PP) was examined in transgenic (Ins1Cre/+;Rosa26-eYFP, GluCreERT2;Rosa26-eYFP) diabetic (STZ-induced) mice. A well-orchestrated multi-component experimental design (bioassay + histological + immunohistochemical + serum/pancreatic biochemical assays), was employed. Authors satisfactorily demonstrated the anti-diabetic action of [P3] PP by comparing findings vs. STZ-control rats. Although the manuscript shows important and relevant data, there’s still room for improvement. Authors are suggested to consider the following when preparing their revised manuscript (v2):

  • Do not forget to describe the meaning of each abbreviation the first time each one is mentioned. If possible, reduce them as much as possible.
  • Figures should be provided with a much higher resolution (>300 dpi) and size; Their footnotes should be self-explanatory without needing to read the main text.
  • Bioassay parameters (Figure 1a-e) could be reported in just one table (including p-values) and other relevant data (if recorded while performing the bioassay) should be included (e.g. time-trend water intake, food efficiency ratio, α/β cell ratio, etc.)
  • When needed, the contribution of this manuscript to the state-of-the-art on this topic should be highlighted (e.g. https://doi.org/10.1016/j.peptides.2022.170923 , https://doi.org/10.1038/s41392-024-02098-3)
  • Check once again your references for non-properly formatted ones (e.g. reference 3= https://doi.org/10.1002/biof.2059).

Author Response

Referee 3

Comment: In this work, the antidiabetic effect of a synthetic polypeptide none-digestible analogue (to pancreatic polypeptide, PP) synthetic polypeptide ([P3]PP) was examined in transgenic (Ins1Cre/+;Rosa26-eYFP, GluCreERT2;Rosa26-eYFP) diabetic (STZ-induced) mice. A well-orchestrated multi-component experimental design (bioassay + histological + immunohistochemical + serum/pancreatic biochemical assays), was employed. Authors satisfactorily demonstrated the anti-diabetic action of (P3)PP by comparing findings vs. STZ-control rats. Although the manuscript shows important and relevant data, there’s still room for improvement. Authors are suggested to consider the following when preparing their revised manuscript (v2):

Author reply: We thank the Reviewer for their positive appraisal of our manuscript and agree that our findings are important and relevant. We are happy to consider all Reviewer suggestions, as detailed below.

Comment: Do not forget to describe the meaning of each abbreviation the first time each one is mentioned. If possible, reduce them as much as possible.

Author reply: The authors have methodically gone through the text and ensured that all abbreviations have now been defined on first use. We have also tried to limit use of abbreviations throughout.

Comment: Figures should be provided with a much higher resolution (>300 dpi) and size; Their footnotes should be self-explanatory without needing to read the main text.

Author reply: The authors are happy to comply with this suggestion and increase resolution of our figures. We have also added more specific and relevant detail to the respective figure legends.

Comment: Bioassay parameters (Figure 1a-e) could be reported in just one table (including p-values) and other relevant data (if recorded while performing the bioassay) should be included (e.g. time-trend water intake, food efficiency ratio, α/β cell ratio, etc.)

Author reply: The authors thank the Reviewer for this suggestion. We did consider this exact approach for data presentation, but since we have two separate animal models, the data did not easily lend itself to being reported in Table format. We would need two separate Tables that seemed a little ‘clunky’ to us. Given that, we thought it best to retain these data in graphical format. We also considered the ability to plot data points in scatter graph form (Figure 1B,E as well as Figure 2A-D) as a more transparent presentation method, which is not possible in tabular form. We hope the Reviewer can understand our approach in this regard.

Comment: When needed, the contribution of this manuscript to the state-of-the-art on this topic should be highlighted (e.g. https://doi.org/10.1016/j.peptides.2022.170923, https://doi.org/10.1038/s41392-024-02098-3)

Author reply: The authors thank the Reviewer for this comment and for the highlighting the two excellent recent, and highly relevant, review articles. To directly address the point raised by Reviewer, we have now added the following sentence to the last paragraph of the Discussion section (Page 16, line 22 and Page 17, lines 1-4), that reads as follows: ‘Overall, PP has not been extensively studied to date with limited knowledge on mechanisms and related effects [37,38], the current study helps to address this issue and confirms important NPY4R-mediated actions on energy regulation and pancreatic endocrine function, with important effects on islet cell lineage that support conservation of normal pancreatic islet morphology’. The reference section has been updated accordingly.

Comment: Check once again your references for non-properly formatted ones (e.g. reference 3= https://doi.org/10.1002/biof.2059).

Author reply: We thank the Reviewer for pointing this out. At time of writing, reference #3 was not yet available in print issue, being listed as ‘Ahead of Print’. Thus, there was no journal bibliographic data available for this citation, but we have now updated accordingly. We have also checked all other references and made any appropriate updates. The authors are thankful for opportunity to improve the accuracy of our paper.

Reviewer 4 Report

Comments and Suggestions for Authors

In this study, Yu and colleagues have exploited two mouse models which provide options for lineage tracing to examine the effects of a long-acting pancreatic polypeptide analogue on islet function and architecture. They have employed multiple low dose administration of streptozotocin as a means to induce diabetes and have studied the effects of prior and sustained treatment with their agonist on glucose homeostasis, metabolic responses and islet architecture. 

Overall, the studies are impressive in scope and the experiments have been designed carefully. The data are presented logically and are considered in a measured way with clear indication of possible limitations as well as emphasis of the positive outcomes. The results reveal very positive effects of the synthetic PP agonist on beta-cell preservation and islet architecture and suggest that this drug might be of value in a clinical context. 

A small number of minor issues might be clarified to good effect:

  1. The control animals appeared to lose weight over the course of the experiment (Fig.1) despite consistent food consumption. Could the authors comment on the reasons?
  2. The study has focussed mainly on islet cells but it would be of interest to understand whether the exocrine pancreas is also affected by the agonist. Were there any changes in pancreas weight or volume occurring parallel with the islet alterations?
  3. The authors argue that the STZ-induced changes in islet architecture are reversed by the PP agonist but it is unclear how this was assessed. Indeed, the images presented appear to suggest that the classical “mantle-core” arrangement of endocrine cells (beta cells in the core) may remain dysregulated after against exposure. Can the authors comment more fully on the extent to which islet morphology was truly restored?
  4. No controls are shown in which the PP agonist is administered in the absence of STZ. Has this been done and, if so, were any effects on islet cell composition or function noted? 

Author Response

Referee 4

Comment: In this study, Yu and colleagues have exploited two mouse models which provide options for lineage tracing to examine the effects of a long-acting pancreatic polypeptide analogue on islet function and architecture. They have employed multiple low dose administration of streptozotocin as a means to induce diabetes and have studied the effects of prior and sustained treatment with their agonist on glucose homeostasis, metabolic responses and islet architecture. Overall, the studies are impressive in scope and the experiments have been designed carefully. The data are presented logically and are considered in a measured way with clear indication of possible limitations as well as emphasis of the positive outcomes. The results reveal very positive effects of the synthetic PP agonist on beta-cell preservation and islet architecture and suggest that this drug might be of value in a clinical context. A small number of minor issues might be clarified to good effect:

Author reply: The authors welcome the positive summary of our paper by this Reviewer. Our comments on the minor points raised are detailed below.

Comment: 1. The control animals appeared to lose weight over the course of the experiment (Fig.1) despite consistent food consumption. Could the authors comment on the reasons?

Author reply: We thank the Reviewer for this comment. We have looked at the data once again, but we are not able to see a loss of weight in the saline treated control animals in Figure 1. Indeed, body weights of healthy control mice were slightly elevated at the end of the study than compared to the first day of experimentation. However, the STZ-diabetic control mice did lose weight over the course of the study, and this effect is characteristic of low-dose STZ and insulin deficiency. To make this clearer within the revised text, the authors have modified the following sentence of the Discussion section (Page 12, lines 13-16), to now read: ‘As expected, multiple low dose STZ administration elevated blood glucose levels in both Ins1Cre/+;Rosa26-eYFP and GluCreERT2;Rosa26-eYFP mice, as a result of specific pancreatic beta-cell destruction, resulting in severely depleted pancreatic insulin stores [23], with this insulin deficiency also leading to anticipated weight loss’.

Comment:  2. The study has focussed mainly on islet cells but it would be of interest to understand whether the exocrine pancreas is also affected by the agonist. Were there any changes in pancreas weight or volume occurring parallel with the islet alterations?

Author reply: The authors agree that our study has focused on islet cells, principally because our transgenic mouse models allow us to accurately investigate islet endocrine cell plasticity. For this reason, we did not measure pancreatic weight or volume. Given that, the authors think it best not to speculate further on these matters within the manuscript to help avoid possible misperceptions. We trust the Reviewer understands our reasoning for this.

Comment: 3. The authors argue that the STZ-induced changes in islet architecture are reversed by the PP agonist, but it is unclear how this was assessed. Indeed, the images presented appear to suggest that the classical “mantle-core” arrangement of endocrine cells (beta cells in the core) may remain dysregulated after against exposure. Can the authors comment more fully on the extent to which islet morphology was truly restored?

Author reply: The Reviewer raises a valid point. He/she is correct that the classical “mantle-core” arrangement of murine pancreatic islet cells was not fully restored by (P3)PP treatment. Our argument for benefits of (P3)PP on islet architecture were chiefly based on observations of enhanced beta-cell area. To address the point raised by the Reviewer, the authors have added the following sentence to the Discussion section of the revised manuscript (Page 15, lines 10-12), that reads as follows: ‘In addition, although beta-cell area was enhanced by (P3)PP treatment in both mouse models, STZ-induced impairment of the classic murine islet ‘alpha-cell mantle/beta-cell core’ was still evident’.

Comment: 4. No controls are shown in which the PP agonist is administered in the absence of STZ. Has this been done and, if so, were any effects on islet cell composition or function noted? 

Author reply: We have not investigated the impact of prolonged treatment with (P3)PP in normal healthy mice because we are primarily interested in the therapeutic actions of this peptide. However,  we agree with the value in conducting such experiments and that is something to consider in the future.  

Round 2

Reviewer 1 Report

Comments and Suggestions for Authors

The authors responded to my comments adequately, and the paper has been improved significantly.